# NOIR: Neural Signal Operated Intelligent Robots for Everyday Activities

**Ruohan Zhang*[1,4], Sharon Lee*[1], Minjune Hwang*[1], Ayano Hiranaka*[2],**
**Chen Wang[1], Wensi Ai[1], Jin Jie Ryan Tan[1], Shreya Gupta[1], Yilun Hao[1],**
**Gabrael Levine[1], Ruohan Gao[1], Anthony Norcia[3], Li Fei-Fei[1,4], Jiajun Wu[1,4]**
*Equally contributed; zharu@stanford.edu
[1]Department of Computer Science, Stanford University
[2]Department of Mechanical Engineering, Stanford University
[3]Department of Psychology, Stanford University
[4]Institute for Human-Centered AI (HAI), Stanford University

**Abstract:** We present Neural Signal Operated Intelligent Robots (NOIR), a general-purpose, intelligent brain-robot interface system that enables humans to command robots to perform everyday activities through brain signals. Through this interface, humans communicate their intended objects of interest and actions to the robots using electroencephalography (EEG). Our novel system demonstrates success in an expansive array of 20 challenging, everyday household activities, including cooking, cleaning, personal care, and entertainment. The effectiveness of the system is improved by its synergistic integration of robot learning algorithms, allowing for NOIR to adapt to individual users and predict their intentions. Our work enhances the way humans interact with robots, replacing traditional channels of interaction with direct, neural communication. Project website: https://noir-corl.github.io/

**Keywords:** Brain-Robot Interface; Human-Robot Interaction

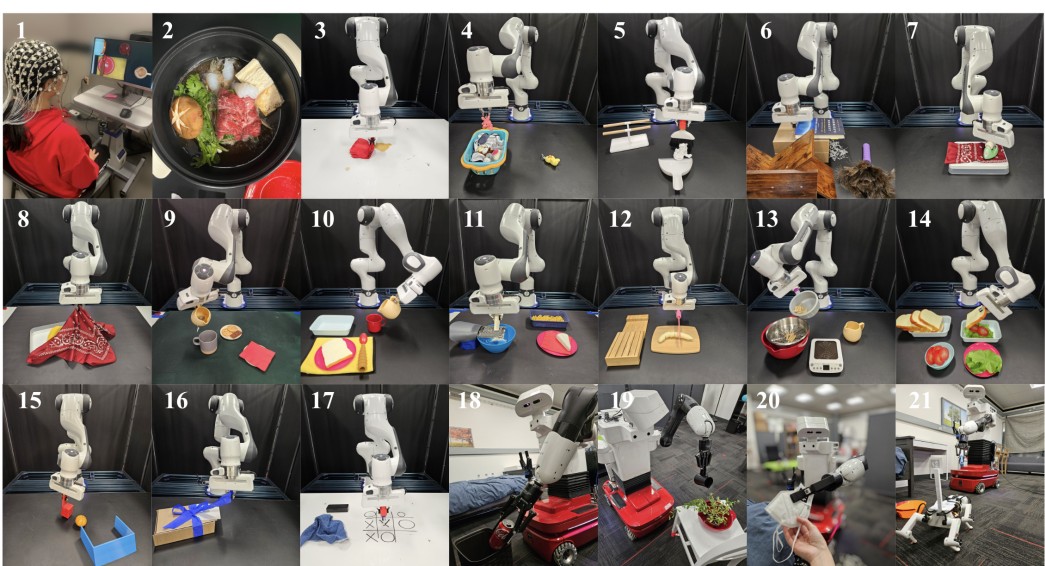

Figure 1: NOIR is a general-purpose brain-robot interface that allows humans to use their brain signals (1) to control robots to perform daily activities, such as making Sukiyaki (2), ironing clothes (7), playing Tic-Tac-Toe with friends (17), and petting a robot dog (21).

7th Conference on Robot Learning (CoRL 2023), Atlanta, USA.

# 1 Introduction

Brain-robot interfaces (BRIs) are a pinnacle achievement in the realm of art, science, and engineering. This aspiration, which features prominently in speculative fiction, innovative artwork, and groundbreaking scientific studies, entails creating robotic systems that operate in perfect synergy with humans. A critical component of such systems is their ability to communicate with humans. In human-robot collaboration and robot learning, humans communicate their intents through actions [1], button presses [2, 3], gaze [4–7], facial expression [8], language [9, 10], etc [11, 12]. However, the prospect of direct communication through neural signals stands out to be the most thrilling but challenging medium.

We present Neural Signal Operated Intelligent Robots (NOIR), a general-purpose, intelligent BRI system with non-invasive electroencephalography (EEG). The primary principle of this system is hierarchical shared autonomy, where humans define high-level goals while the robot actualizes the goals through the execution of low-level motor commands. Taking advantage of the progress in neuroscience, robotics, and machine learning, our system distinguishes itself by extending beyond previous attempts to make the following contributions.

First, NOIR is *general-purpose* in its diversity of tasks and accessibility. We show that humans can accomplish an expansive array of 20 daily everyday activities, in contrast to existing BRI systems that are typically specialized at one or a few tasks or exist solely in simulation [13–22]. Additionally, the system can be used by the general population, with a minimum amount of training.

Second, the "I" in NOIR means that our robots are *intelligent* and adaptive. The robots are equipped with a library of diverse skills, allowing them to perform low-level actions without dense human supervision. Human behavioral goals can naturally be communicated, interpreted, and executed by the robots with *parameterized primitive skills*, such as Pick(obj-A) or MoveTo(x,y). Additionally, our robots are capable of learning human intended goals during their collaboration. We show that by leveraging the recent progress in foundation models, we can make such a system more adaptive with limited data. We show that this can significantly increase the efficiency of the system.

The key technical contributions of NOIR include a *modular* neural signal decoding pipeline for human intentions. Decoding human intended goals (e.g., "pick up the mug from the handle") from neural signals is extremely challenging. We decompose human intention into three components: *What* object to manipulate, *How* to interact with the object, and *Where* to interact, and show that such signals can be decoded from different types of neural data. These decomposed signals naturally correspond to parameterized robot skills and can be communicated effectively to the robots.

In 20 household activities involving tabletop or mobile manipulations, three human subjects successfully used our system to accomplish these tasks with their brain signals. We demonstrate that few-shot robot learning from humans can significantly improve the efficiency of our system. This approach to building intelligent robotic systems, which utilizes human brain signals for collaboration, holds immense potential for the development of critical assistive technologies for individuals with or without disabilities and to improve the quality of their life.

# 2 Brain-Robot Interface (BRI): Background

Since Hans Berger's discovery of EEG in 1924, several types of devices have been developed to record human brain signals. We chose non-invasive, saline-based EEG due to its cost and accessibility to the general population, signal-to-noise ratio, temporal and spatial resolutions, and types of signals that can be decoded (see Appendix 2). EEG captures the spontaneous electrical activity of the brain using electrodes placed on the scalp. EEG-based BRI has been applied to prosthetics, wheelchairs, as well as navigation and manipulation robots. For comprehensive reviews, see [22–25]. We utilize two types of EEG signals that are frequently employed in BRI, namely, steady-state visually evoked potential (SSVEP) and motor imagery (MI).

SSVEP is the brain's exogenous response to periodic external visual stimulus [26], wherein the brain generates periodic electrical activity at the same frequency as flickering visual stimulus. The appli-

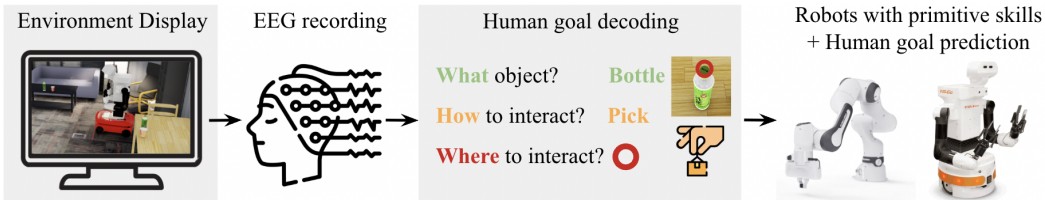

Figure 2: NOIR has two components, a modular pipeline for decoding goals from human brain signals, and a robotic system with a library of primitive skills. The robots possess the ability to learn to predict human intended goals hence reducing the human effort required for decoding.

cation of SSVEP in assistive robotics often involves the usage of flickering LED lights physically affixed to different objects [27, 28]. Attending to an object (and its attached LED light) will increase the EEG response at that stimulus frequency, allowing the object's identity to be inferred. Inspired by prior work [15], our system utilizes computer vision techniques to detect and segment objects, attach virtual flickering masks to each object, and display them to the participants for selection.

Motor Imagery (MI) differs from SSVEP due to its endogenous nature, requiring individuals to mentally simulate specific actions, such as imagining oneself manipulating an object. The decoded signals can be used to indicate a human's intended way of interacting with the object. This approach is widely used for rehabilitation, and for navigation tasks [29] in BRI systems. This approach often suffers from low decoding accuracy [22].

Much existing BRI research focuses on the fundamental problem of brain signal decoding, while several existing studies focus on how to make robots more intelligent and adaptive [13–17, 30]. Inspired by this line of work, we leverage few-shot policy learning algorithms to enable robots to learn human preferences and goals. This minimizes the necessity for extensive brain signal decoding, thereby streamlining the interaction process and enhancing overall efficiency.

Our study is grounded in substantial advancements in both the field of brain signal decoding and robot learning. Currently, many existing BRI systems target only one or a few specific tasks. To the best of our knowledge, no previous work has presented an intelligent, versatile system capable of successfully executing a wide range of complex tasks, as demonstrated in our study.

## 3 The NOIR System

The challenges we try to tackle are: 1) How do we build a general-purpose BRI system that works for a variety of tasks? 2) How do we decode relevant communication signals from human brains? 3) How do we make robots more intelligent and adaptive for more efficient collaboration? An overview of our system is shown in Fig. 2. Humans act as planning agents to perceive, plan, and communicate behavioral goals to the robot, while robots use pre-defined primitive skills to achieve these goals.

The overarching goal of building a general-purpose BRI system is achieved by synergistically integrating two designs together. First, we propose a novel *modular* brain decoding pipeline for human intentions, in which the human intended goal is decomposed into three components: what, how, and where (Sec. 3.1). Second, we equip the robots with a library of parameterized primitive skills to accomplish human-specified goals (Sec. 3.2). This design enables humans and robots to collaborate to accomplish a variety of challenging, long-horizon everyday tasks. At last, we show a key feature of NOIR to allow robots to act more efficiently and to be capable of adapting to individual users, we adopt few-shot imitation learning from humans (Sec. 3.3).

### 3.1 The brain: A modular decoding pipeline

We hypothesize that the key to building a general-purpose EEG decoding system is modularization. Decoding complete behavioral goals (e.g., in the form of natural language) is only feasible with expensive devices like fMRI, and with many hours of training data for each individual [31]. As shown in Fig. 3, we decompose human intention into three components: (a) *What* object to manipulate; (b) *How* to interact with the object; (c) *Where* to interact. The decoding of specific user intents from

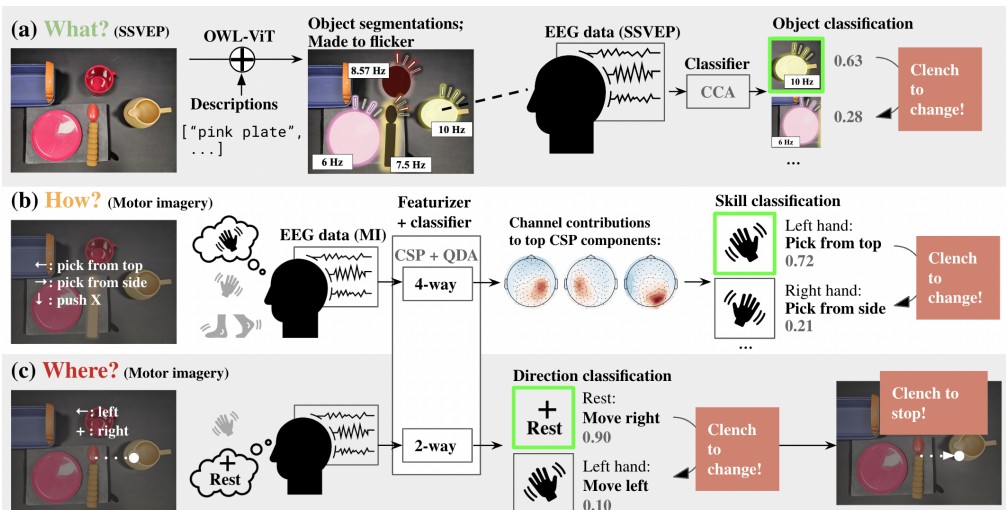

Figure 3: A modular pipeline for decoding human intended goals from EEG signals: (a) *What* object to manipulate, decoded from SSVEP signals using CCA classifiers; (b) *How* to interact with the object, decoded from MI signals using CSP+QDA algorithms; (c) *Where* to interact, decoded from MI signals. A safety mechanism that captures muscle tension from jaw clench is used to confirm or reject decoding results.

EEG signals is challenging but can be done with steady-state visually evoked potential and motor imagery, as introduced in Sec. 2. For brevity, details of decoding algorithms are in Appendix 6.

**Selecting objects with steady-state visually evoked potential (SSVEP).** Upon showing the task set-up on a screen, we first infer the user's intended object. We make objects on the screen flicker with different frequencies (Fig. 3a), which, when focused on by the user, evokes SSVEP [26]. By identifying which frequency is stronger in the EEG data, we may infer the frequency of the flickering visual stimulus, and hence the object that the user focuses on. We apply modern computer vision techniques to circumvent the problem of having to physically attach LED lights [27, 28]. Specifically, we use the foundation model OWL-ViT [32] to detect and track objects, which takes in an image and object descriptions and outputs object segmentation masks. By overlaying each mask of different flickering frequencies ($6Hz$, $7.5Hz$, $8.57Hz$, and $10Hz$ [33, 34]), and having the user focus on the desired object for 10 seconds, we are able to identify the attended object.

We use only the signals from the visual cortex (Appendix 6) and preprocess the data with a notch filter. We then use Canonical Correlation Analysis (CCA) for classification [35]. We create a Canonical Reference Signal (CRS), which is a set of $\sin$ and $\cos$ waves, for each of our frequencies and their harmonics. We then use CCA to calculate the frequency whose CRS has the highest correlation with the EEG signal, and identify the object that was made to flicker at that frequency.

**Selecting skill and parameters with motor imagery (MI).** The user then chooses a skill and its parameters. We frame this as a $k$-way ($k \leq 4$) MI classification problem, where we aim to decode which of the $k$ pre-decided actions the user was imagining. Unlike SSVEP, a small amount of calibration data (10-min) is required due to the distinct nature of each user's MI signals. The four classes are: `Left Hand`, `Right Hand`, `Legs`, and `Rest`; the class names describe the body parts that users imagine using to execute some skills (e.g. pushing a pedal with feet). Upon being presented with the list of $k$ skill options, we record a 5-second EEG signal, and use a classifier trained on the calibration data. The user then guides a cursor on the screen to the appropriate location for executing the skill. To move the cursor along the $x$ axis, the user is prompted to imagine moving their `Left` hand for leftward cursor movement. We record another five seconds of data and utilize a 2-way classifier. This process is repeated for $x$, $y$, and $z$ axes.

For decoding, we use only EEG channels around the brain areas related to motor imagery (Appendix 6). The data is band-pass-filtered between $8Hz$ and $30Hz$ to include $\mu$-band and $\beta$-band frequency ranges correlated with MI activity [36]. The classification algorithm is based on the common spatial

pattern (CSP) [37–40] algorithm and quadratic discriminant analysis (QDA). Due to its simplicity, CSP+QDA is explainable and amenable to small training datasets. Contour maps of electrode contributions to the top few CSP-space principal components are shown in the middle row of Fig. 3. There are distinct concentrations around the right and left motor areas, as well as the visual cortex (which correlates with the `Rest` class).

**Confirming or interrupting with muscle tension.** Safety is critical in BRI due to noisy decoding. We follow a common practice and collect electrical signals generated from facial muscle tension (Electromyography, or EMG). This signal appears when users frown or clench their jaws, indicating a negative response. This signal is strong with near-perfect decoding accuracy, and thus we use it to confirm or reject object, skill, or parameter selections. With a pre-determined threshold value obtained through the calibration stage, we can reliably detect muscle tension from 500-ms windows.

## 3.2 The robot: Parameterized primitive skills

Our robots must be able to solve a diverse set of manipulation tasks under the guidance of humans, which can be achieved by equipping them with a set of parameterized primitive skills. The benefits of using these skills are that they can be combined and reused across tasks. Moreover, these skills are intuitive to humans. Since skill-augmented robots have shown promising results in solving long-horizon tasks, we follow recent works in robotics with parameterized skills [41–52], and augment the action space of our robots with a set of primitive skills and their parameters. Neither the human nor the agent requires knowledge of the underlying control mechanism for these skills, thus the skills can be implemented in any method as long as they are robust and adaptive to various tasks.

We use two robots in our experiment: A Franka Emika Panda arm for tabletop manipulation tasks, and a PAL Tiago robot for mobile manipulation tasks (see Appendix for hardware details). Skills for the Franka robot use the operational space pose controller (OSC) [53] from the Deoxys API [54]. For example, `Reaching` skill trajectories are generated by numerical 3D trajectory interpolation conditioned on the current robot end-effector 6D pose and target pose. Then OSC controls the robot to reach the waypoints along the trajectory orderly. The Tiago robot's navigation skill is implemented using the ROS MoveBase package, while all other skills are implemented using MoveIt motion planning framework [55]. A complete list of skills for both robots is in Appendix 3. Later, we will show that humans and robots can work together using these skills to solve all the tasks.

## 3.3 Leveraging robot learning for efficient BRI

The modular decoding pipeline and the primitive skill library lay the foundation for NOIR. However, the efficiency of such a system can be further improved. During the collaboration, the robots should learn the user's object, skill, and parameter selection preferences, hence in future trials, the robot can predict users' intended goals and be more autonomous, hence reducing the effort required for decoding. Learning and generalization are required since the location, pose, arrangement, and instance of the objects could differ from trial to trial. Meanwhile, the learning algorithms should be sample-efficient since human data is expensive to collect.

**Retrieval-based few-shot object and skill selection.** In NOIR, human effort can be reduced if the robot intelligently learns to propose appropriate object-skill selections for a given state in the task. Inspired by retrieval-based imitation learning [56–58], our proposed method learns a latent state representation from observed states. Given a new state observation, it finds the most similar state in the latent space and the corresponding action. Our method is shown in Fig. 4. During task execution, we record data points that consist of images and the object-skill pairs selected by the human. The images are first encoded by a pre-trained R3M model [59] to extract useful features for robot manipulation tasks, and are then passed through several trainable, fully-connected layers. These layers are trained using contrastive learning with a triplet loss[60] that encourages the images with the same object-skill label to be embedded closer in the latent space. The learned image embeddings and object-skill labels are stored in the memory. During test time, the model retrieves the nearest data point in the latent space and suggests the object-action pair associated with that data point to the human. Details of the algorithm can be found in Appendix 7.1.

**One-shot skill parameter learning.** Parameter selection requires a lot of human effort as it needs precise cursor manipulation through MI. To reduce human effort, we propose a learning algorithm

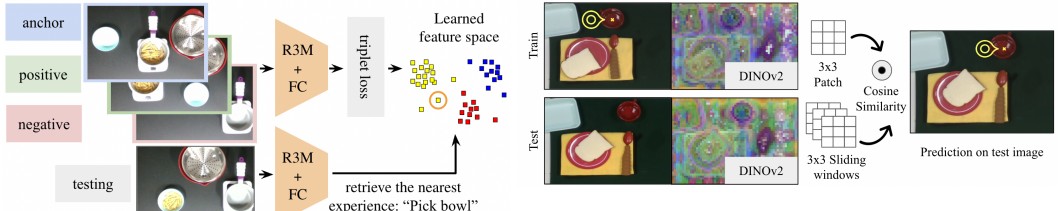

Figure 4: Left: Retrieval-based few-shot object and skill selection model. The model learns a latent representation for observations. Given a new observation, it finds the most relevant experience in the memory and selects the corresponding skill and object. Right: One-shot skill parameter learning algorithm, which finds a semantically corresponding point in the test image given a reference point in the training image. The feature visualization shows 3 of the 768 DINOv2 tokens used.

for predicting parameters given an object-skill pair as an initial point for cursor control. Assuming that the user has once successfully pinpointed the precise key point to pick a mug's handle, does this parameter need to be specified again in the future? Recent advancement in foundation models such as DINOv2 [61] allows us to find corresponding semantic key points, eliminating the need for parameter re-specification. Compared to previous works, our algorithm is one-shot [62–66] and predicts specific 2D points instead of semantic segments [67, 68]. As shown in Fig. 4, given a training image ($360 \times 240$) and parameter choice $(x, y)$, we predict the semantically corresponding point in the test images, in which positions, orientations, instances of the target object, and contexts may vary. We utilize a pre-trained DINOv2 model to obtain semantic features [61]. We input both train and test images into the model and generate 768 patch tokens, each as a pixel-wise feature map of dimension $75 \times 100$. We then extract a $3 \times 3$ patch centered around the provided training parameter and search for a matching feature in the test image, using cosine similarity as the distance metric. Details of this algorithm can be found in Appendix 7.2.

## 4  Experiments

**Tasks.** NOIR can greatly benefit those who require assistance with everyday activities. We select tasks from the BEHAVIOR benchmark [69] and Activities of Daily Living [70] to capture actual human needs. The tasks are shown in Fig. 1, and consist of 16 tabletop tasks and four mobile manipulation tasks. The tasks encompass various categories, including eight meal preparation tasks, six cleaning tasks, three personal care tasks, and three entertainment tasks. For systematic evaluation of task success, we provide formal definitions of these activities in the BDDL language format [69, 71], which specifies the initial and goal conditions of a task using first-order logic. Task definitions and figures can be found in Appendix 4.

**Procedure.** The human study conducted has received approval from Institutional Review Board. Three healthy human participants (2 male, 1 female) performed all 15 Franka tasks. Sukiyaki, four Tiago tasks, and learning tasks are performed by one user. We use the EGI NetStation EEG system, which is completely non-invasive, making almost everyone an ideal subject. Before experiments, users are familiarized with task definitions and system interfaces. During the experiment, users stay in an isolated room, remain stationary, watch the robot on a screen, and solely rely on brain signals to communicate with the robots (more details about the procedure can be found in Appendix 5).

## 5  Results

We seek to provide answers to the following questions through extensive evaluation: 1) Is NOIR truly general-purpose, in that it allows all of our human subjects to accomplish the diverse set of everyday tasks we have proposed? 2) Does our decoding pipeline provide accurate decoding results? 3) Does our proposed robot learning and intention prediction algorithm improve NOIR's efficiency?

**System performance.** Table 1 summarizes the performance based on two metrics: the number of attempts until success and the time to complete the task in successful trials. When the participant reached an unrecoverable state in task execution, we reset the environment and the participant re-attempted the task from the beginning. Task horizons (number of primitive skills executed) are

| Task | WipeSpill | CollectToy | SweepTrash | CleanBook | IronCloth | OpenBasket | PourTea | SetTable | GrateCheese | CutBanana |
|---|---|---|---|---|---|---|---|---|---|---|
| Task horizon | 4.33 | 7.67 | 5.67 | 7.00 | 4.67 | 5.33 | 4.00 | 8.33 | 7.00 | 5.33 |
| # Attempts | 1.00 | 1.33 | 2.33 | 3.33 | 2.33 | 1.67 | 1.67 | 5.67 | 1.33 | 1.67 |
| Time (min) | 14.74 | 25.24 | 20.59 | 27.73 | 16.95 | 15.90 | 13.53 | 20.91 | 24.98 | 17.68 |
| Human time (%) | 79.02 | 83.97 | 82.34 | 80.00 | 79.56 | 82.03 | 83.15 | 81.15 | 81.79 | 81.21 |

| Task | CookPasta | Sandwich | Hockey | OpenGift | TicTacToe | Sukiyaki | TrashDisposal | CovidCare | WaterPlant | PetDog |
|---|---|---|---|---|---|---|---|---|---|---|
| Task horizon | 8.33 | 9.00 | 5.00 | 7.00 | 14.33 | 13.00 | 8.00 | 8.00 | 4.00 | 6.00 |
| # Attempts | 1.67 | 1.67 | 1.33 | 2.67 | 2.00 | 1.00 | 1.00 | 1.00 | 1.00 | 1.00 |
| Time (min) | 30.06 | 27.87 | 15.83 | 23.57 | 43.08 | 43.45 | 7.25 | 8.80 | 3.00 | 4.58 |
| Human time (%) | 83.26 | 82.71 | 82.00 | 79.90 | 80.54 | 84.85 | 55.32 | 62.29 | 87.41 | 87.53 |

Table 1: NOIR system performance. Task horizon is the average number of primitive skills executed. # attempts indicate the average number of attempts until the first success (1 means success on the first attempt). Time indicates the task completion time in successful trials. Human time is the percentage of the total time spent by human users, this includes decision-making time and decoding time. With only a few attempts, all users can accomplish these challenging tasks.

included as a reference. Although these tasks are long-horizon and challenging, NOIR shows very encouraging results: on average, tasks can be completed with only 1.83 attempts. The reason for task failures is human errors in skill and parameter selection, i.e., the users pick the wrong skills or parameters, which leads to non-recoverable states and needs manual resets. Decoding errors or robot execution errors are avoided thanks to our safety mechanism with confirmation and interruption. Although our primitive skill library is limited, human users find novel usage of these skills to solve tasks creatively. Hence we observe emerging capabilities such as extrinsic dexterity. For example, in task `CleanBook` (Fig. 1.6), Franka's `Pick` skill is not designed to grasp a book from the table, but users learn to push the book towards the edge of the table and grasp it from the side. In `CutBanana` (Fig. 1.12), users utilize `Push` skill to cut. The average task completion time is 20.29 minutes. Note that the time humans spent on decision-making and decoding is relatively long (80% of total time), partially due to the safety mechanism. Later, we will show that our proposed robot learning algorithms can address this issue effectively.

**Decoding accuracy.** A key to our system's success is the accuracy in decoding brain signals. Table 2 summarizes the decoding accuracy of different stages. We find that CCA on SSVEP produces a high accuracy of $81.2\%$, meaning that object selection is mostly accurate. As for CSP + QDA on MI for parameter selection, the 2-way classification model performs at $73.9\%$ accuracy, which is consistent with current literature [36]. The 4-way skill-selection classification models perform at about $42.2\%$ accuracy. Though this may not seem high, it is competitive considering inconsistencies attributed to long task duration (hence the discrepancy between calibration and task-time accuracies). Our calibration time is only 10 minutes, which is significantly shorter compared to the duration of typical MI calibration and training sessions by several orders of magnitude [21]. More calibration provides more data for training more robust classifiers, and allows human users to practice more which typically yields stronger brain signals. Overall, the decoding accuracy is satisfactory, and with the safety mechanism, there has been no instance of task failure caused by incorrect decoding.

**Object and skill selection results.** We then answer the third question: Does our proposed robot learning algorithm improve NOIR's efficiency? First, we evaluate object and skill selection learning. We collect a dataset offline with 15 training samples for each object-skill pair in `MakePasta` task. Given an image, a prediction is considered correct if both the object and the skill are predicted correctly. Results are shown in Table 3. While a simple image classification model using ResNet [72] achieves an average accuracy of 0.31, our method with a pre-trained ResNet backbone achieves significantly higher accuracy at 0.73, highlighting the importance of contrastive learning and retrieval-based learning. Using R3M as the feature extractor further improves the performance to 0.94. The generalization ability of the algorithm is tested on the same `MakePasta` task. For instance-level generalization, 20 different types of pasta are used; for context generalization, we randomly select and place 20 task-irrelevant objects in the background. Results are shown in Table 3. In all variations, our model achieves accuracy over 93%, meaning that the human can skip the skill and object selection 93% of the time, significantly reducing their time and effort. We further test our algorithm during actual task execution (Fig. 5). A human user completes the task with and without object-skill prediction two times each. With object and skill learning, the average time required for each object-skill selection is reduced by 60% from 45.7 to 18.1 seconds. More details about the experiments and visualization of learned representation can be found in Appendix 7.1.

| Decoding Stage | Signal | Technique | Calibration Acc. | Task-Time Acc. |
|---|---|---|---|---|
| Object selection (What?) | SSVEP | CCA (4-way) | - | 0.812 |
| Skill selection (How?) | MI | CSP + QDA (4-way) | 0.580 | 0.422 |
| Parameter selection (Where?) | MI | CSP + QDA (2-way) | 0.882 | 0.739 |
| Confirmation / interruption | EMG | Thresholding (2-way) | 1.0 | 1.0 |

Table 2: Decoding accuracy at different stages of the experiment.

**One-shot parameter learning results.** First, using our pre-collected dataset (see Appendix 7.2), we compare our algorithm against multiple baselines. The MSE values of the predictions are shown in Table 4. *Random sample* shows the average error when randomly predicting points in the 2D space. *Sample on objects* randomly predicts a point on objects and not on the background; the object masks here are detected with the Segment Anything Model (SAM) [73]. For *Pixel similarity*, we employ the cosine similarity and sliding window techniques used in our algorithm, but on raw images without using DINOv2 features. All of the baselines are drastically outperformed by our algorithm. Second, our one-shot learning method demonstrates robust generalization capability, as tested on the respective dataset; table 4 presents the results. The low prediction error means that users spend much less effort in controlling the cursor to move to the desired position. We further demonstrate the effectiveness of the parameter learning algorithm in actual task execution for `SetTable`, quantified in terms of saved human effort in controlling the cursor movement (Fig. 5). Without learning, the cursor starts at the chosen object or the center of the screen. The predicted result is used as the starting location for cursor control which led to a considerable decrease in cursor movement, with the mean distance reduced by 41%. These findings highlight the potential of parameter learning in improving efficiency and reducing human effort. More results and visualizations can be found in Appendix 7.2.

| Method | Acc.↑ | Generalization | Acc.↑ |
|---|---|---|---|
| Random | 0.12±0.02 | Position | 0.95±0.04 |
| Classfication (ResNet) | 0.31±0.11 | Pose | 0.94±0.04 |
| Ours (ResNet) | 0.73±0.09 | Instance | 0.93±0.02 |
| Ours (R3M) | 0.94±0.04 | Context | 0.98±0.02 |

Table 3: Object-skill learning results. Our method is highly accurate and robust.

| Method | MSE↓ | Generalization | MSE↓ |
|---|---|---|---|
| Random sample | 175.8±29.7 | Position | 5.6±6.0 |
| Sample on objects | 137.2±55.7 | Orientation | 12.0±11.7 |
| Pixel similarity | 45.9±50.1 | Instance | 16.4±22.2 |
| Ours | 15.8±23.8 | Context | 26.8±62.5 |

Table 4: One-shot parameter learning results. Our method is highly accurate and generalizes well.

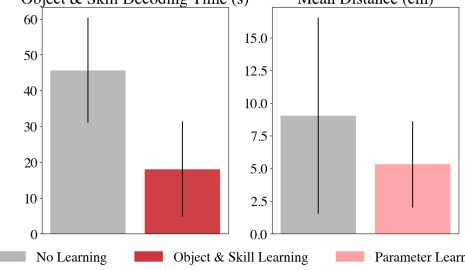

Figure 5: Left: Object and skill selection learning reduces the decoding time by 60%. Right: Parameter learning decreases cursor movement distance by 41%.

## 6 Conclusion, Limitations, and Ethical Concerns

In this work, we presented a general-purpose, intelligent BRI system that allows human users to control a robot to accomplish a diverse, challenging set of real-world activities using brain signals. NOIR enables human intention prediction through few-shot learning, thereby facilitating a more efficient collaborative interaction. NOIR holds a significant potential to augment human capabilities and enable critical assistive technology for individuals who require everyday support.

NOIR represents a pioneering effort in the field, unveiling potential opportunities while simultaneously raising questions about its limitations and potential ethical risks which we address in Appendix 1. The decoding speed, as it currently stands, restricts tasks to those devoid of time-sensitive interactions. However, advancements in the field of neural signal decoding hold promise for alleviating this concern. Furthermore, the compilation of a comprehensive library of primitive skills presents a long-term challenge in robotics, necessitating additional exploration and development. Nonetheless, we maintain that once a robust set of skills is successfully established, human users will indeed be capable of applying these existing skills to complete new tasks.

**Acknowledgments**

The work is in part supported by NSF CCRI #2120095, ONR MURI N00014-22-1-2740, N00014-23-1-2355, N00014-21-1-2801, AFOSR YIP FA9550-23-1-0127, the Stanford Institute for Human-Centered AI (HAI), Amazon, Salesforce, and JPMC.

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

# Appendix 1: Questions and Answers about NOIR

**Q (Safety)** : Is EEG safe to use? Are there any potential risks or side effects of using the EEG for extended periods of time?

**A** : EEG devices are generally safe with no known side effects and risks, especially when compared to invasive devices like implants. We use saline solution to lower electrical impedance and improve conductance. The solution could cause minor skin irritation when the net is used for extended periods of time, hence we mix the solution with baby shampoo to mitigate this.

**Q (Safety)** : How does the system ensure user safety, particularly in the context of real-world tasks with varying environments and unpredictable events?

**A** : We implement an EEG-controlled safety mechanism to confirm or interrupt robot actions with muscle tension, as decoded through clenching. Nevertheless, it is important to note that the current implementation entails a 500ms delay when interrupting robot actions which might lead to a potential risk in more dynamic tasks. With more training data using a shorter decoding window, the issue can be potentially mitigated.

**Q (Universality)** : Can EEG / NOIR be applied to different people? Given that the paper has only been tested on three human subjects, how can the authors justify the generalizability of the findings?

**A** : The EEG device employed in our research is versatile, catering to both adults and children as young as five years old. Accompanied by SensorNets of varying sizes, the device ensures compatibility with different head dimensions. Our decoding methods have been thoughtfully designed with diversity and inclusion in mind, drawing upon two prominent EEG signals: steady-state visually evoked potential and motor imagery. These signals have exhibited efficacy across a wide range of individuals. However, it is important to acknowledge that the interface of our system, NOIR, is exclusively visual in nature, rendering it unsuitable for individuals with severe visual impairments.

**Q (Portability)** : Can EEG be used outside the lab?

**A** : While mobile EEG devices offer portability, it is worth noting that they often exhibit a comparatively much lower signal-to-noise ratio. Various sources contribute to the noise present in EEG signals, including muscle movements, eye movements, power lines, and interference from other devices. These sources of noise exist in and outside of the lab; consequently, though we've chosen to implement robust decoding techniques based on classical statistics, more robust further filtering techniques to mitigate these unwanted artifacts and extract meaningful information accurately are needed for greater success in more chaotic environments.

**Q (Privacy)** : How does the system differentiate between intentional brain signals for task execution and other unrelated brain activity? How will you address potential issues of privacy and security?

**A** : The decoding algorithms employed in our study were purposefully engineered to exclusively capture task-relevant signals, ensuring the exclusion of any extraneous information. Adhering to the principles of data privacy and in compliance with the guidelines set by the Institutional Review Board (IRB) for human research, the data collected from participants during calibration and experimental sessions were promptly deleted following the conclusion of each experiment. Only the decoded signals, stripped of any identifying information, were retained for further analysis.

**Q (Scalability)** : How scalable is the robotics system? Can it be easily adapted to different robot platforms or expanded to accommodate a broader range of tasks beyond the 20 household activities tested?

**A** : Within the context of our study, two notable constraints are the speed of decoding and the availability of primitive skills. The former restricts the range of tasks to those that do not involve time-sensitive and dynamic interactions. However, the advancement in decoding accuracy and the reduction of the decoding window duration may eventually address this limitation. These improvements can potentially be achieved through the utilization of larger training datasets and the implementation of machine-learning-based decoding models, leveraging the high temporal resolution offered by EEG.

| Property | gel-based EEG | dry EEG | MEG | fMRI | fNIRS | implant |
|---|---|---|---|---|---|---|
| Invasive? | No | No | No | No | No | Yes |
| Cost | similar | lower | higher | higher | varies | higher |
| Universality | similar | better | similar | similar | similar | worse |
| Setup time | longer | shorter | similar | similar | longer | longer |
| Signal-to-noise ratio | better | worse | - | - | - | better |
| Temporal resolution | similar | lower | similar | lower | lower | - |
| Spatial resolution | similar | lower | higher | higher | higher | - |

Table 5: A comparison between brain recording devices, using our saline-based EEG device as the baseline. Note that the comparison is based on the average products that are available on the market for research, and does not account for specialized or customized devices. Universality considers whether the device can be used by the general population. For signal-to-noise ratio, MEG, fMRI, and fNIRS record different types of neural signals which are not directly comparable to EEG. For implants, the temporal and spatial resolution largely depends on the particular type of implant device used.

The development of a comprehensive library of primitive skills stands as a long-term objective in the field of robotics research. This entails creating a repertoire of fundamental abilities that can be adapted and combined to address new tasks. Additionally, our findings indicate that human users possess the ability to innovate and devise novel applications of existing skills to accomplish tasks, akin to the way humans employ tools.

**Q (Potential impact)** : How exactly do both individuals with and without disabilities benefit from this BRI system?

**A** : The potential applications of systems like NOIR in the future are vast and diverse. One significant area where these systems can have a profound impact is in assisting individuals with disabilities, particularly those with mobility-related impairments. By enabling these individuals to accomplish Activities of Daily Living and Instrumental Activities of Daily Living [70] tasks, such systems can greatly enhance their independence and overall quality of life. Currently, individuals without disabilities may initially find the BRI pipeline to have a learning curve, resulting in inefficiencies compared to their own performance in daily activities in their first few attempts. However, robot learning methods hold the promise of addressing these inefficiencies over time, and enable robots to help their users when needed.

## Appendix 2: Comparison between Different Brain Recording Devices

We use the EGI NetStation EEG system which uses rapid application 128-channel saline-based EGI SensorNets. Here we justify our choice of using non-invasive, saline-based EEG as the recording device for brain signals. A comparison of different brain reading devices (gel-based EEG, dry EEG, MEG, fMRI, fNIRS, implant) and their advantages and disadvantages are shown in Table 5, using our device as the baseline. Two noticeable alternatives are functional magnetic resonance imaging (fMRI) and invasive implants. fMRI measures the small changes in blood flow that occur with brain activity, which has a very high spatial resolution hence fine-grained information such as object categories and language [31] can be decoded from it. But fMRI suffers from low temporal resolution, and the recording device is extremely costly and cannot be used in daily scenarios. Brain implants have a very good signal-to-noise ratio and have great potential. However, the main concern is that it requires surgery to be applied, and health-related risks are not negligible.

## Appendix 3: System Setup

**Robot platform.** The robot we use in our tabletop manipulation task is a standard Franka Emika robot arm with three RealSense cameras. For mobile manipulation, we use a Tiago++ model from PAL Robotics, with an omnidirectional base, two 7-degrees-of-freedom arms with parallel-yaw grippers, a 1-degree-of-freedom prismatic torso, two SICK LiDAR sensors (back and front of the base),

| Robot | Skill | Parameters |
|---|---|---|
| Franka | Reaching | 6D goal pose in world |
| Franka | Picking | 3D world pos to pick, gripper orientation (choose from 4) |
| Franka | Placing | 3D world pos to place, gripper orientation (choose from 3) |
| Franka | Pushing | 3D world pos to start pushing, axis of motion (choose from 3) |
| Franka | Wiping | 3D world pos to start wiping |
| Franka | Drawing | 3D world pos |
| Franka | Pouring | 3D world pos, gripper orientation (choose from 3) |
| Franka | Pulling | 3D world pos, gripper orientation (choose from 2), pull direction (choose from 2) |
| Franka | Grating | 3D world pos |
| Tiago | Navigating | ID of pre-defined positions and poses |
| Tiago | Picking | ID of the object |
| Tiago | Placing | ID of the object |
| Tiago | Pouring | ID of the object |
| Tiago | Dropping | ID of object to drop the grasped object by |

Table 6: Parameterized primitive skills for Franka and Tiago robots.

and an ASUS Xtion RGB-D camera mounted on the robot's head, which can be controlled in yaw and pitch. All sensors and actuators are connected through the Robot Operating System, ROS [74]. The code runs on a laptop with an Nvidia GTX 1070 that sends the commands to the onboard robot computer to be executed.

**Primitive skills list.** A list of primitive skills along with their parameters can be found in Table 6, eight for Franka (16 tasks) and five for Tiago (four tasks). Human users can accomplish all 20 tasks, which are long-horizon and challenging, using these skills.

## Appendix 4: Task Definitions

For systematic evaluation of task success, we provide formal definitions of our tasks in the format of BEHAVIOR Domain Definition Language (BDDL) language [69, 71]. BDDL is a predicate logic-based language that establishes a symbolic state representation built on predefined, meaningful predicates grounded in physical states [71]. Each task is defined in BDDL as an initial and goal condition parametrizing sets of possible initial states and satisfactory goal states, as shown in the figures at the end of the appendix. Compared to scene- or pose-specific definitions which are too restricted, BDDL is more intuitive to humans while providing concrete evaluation metrics for measuring task success.

## Appendix 5: Experimental Procedure

**EEG device preparation.** In our experiments, we use the 128-channel HydroCel Geodesic SensorNet from Magstim EGI, which has sponge tips in its electrode channels. Prior to experiments, the EEG net is soaked in a solution containing dissolved conductive salt (Potassium Chloride) and baby shampoo for 15 minutes. After the soaking, the net is worn by the experiment subject, and an impedance check is done. This impedance check entails ensuring that the impedance of each channel electrode is $\leq 50.0$ k$\Omega$, by using a syringe to add more conductive fluid between the electrodes and the scalp. We then carefully put on a shower cap to minimize the drying of conductive fluid over the course of the experiment.

**Instructions to subjects.** Before commencing the experiments, subjects are given instructions on how to execute the SSVEP, MI, and muscle tension (jaw-clenching) tasks. For SSVEP, they are instructed to simply focus on the flickering object of interest without getting distracted by the other objects on the screen. For MI, similar to datasets such as BCI Competition 2003 [75], and as per extensive literature review [76], we instruct subjects to either imagine continually bending their hands at the wrist (wrist dorsiflexion) or squeezing a ball for the hand actions ("Left", "Right"), and to imagine depressing a pedal with both feet (feet dorsiflexion) for the "Legs" action. For the "Rest" class, as is common practice in EEG experiments in general, we instruct users to focus on a fixation cross displayed on the screen. Subjects were told to stick with their actions of choice throughout

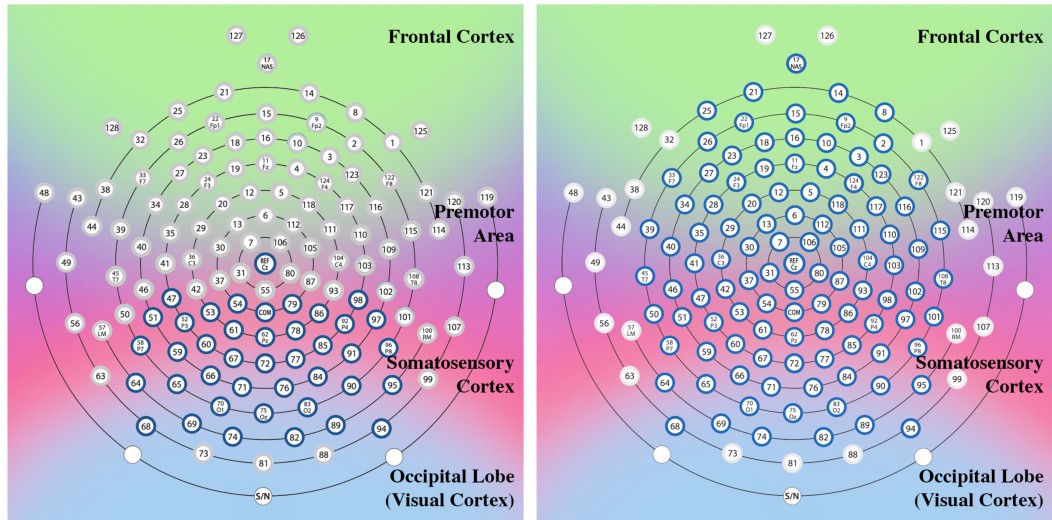

Figure 6: Map of relevant electrodes we use during SSVEP (Left) and Motor Imagery (Right).

the experiment, for consistency. For muscle tension, subjects were told to simply clench their jaw without too much or too little effort.

**Interface.** For SSVEP, subjects are told in writing on the screen to focus on the object of interest. Thereafter, a scene image of the objects with flickering masks overlaid on each object is presented, and we immediately begin recording the EEG Data over this period of time. For MI, the cues are different during calibration and task-time. During calibration, subjects are presented with a warning symbol (.) on screen for 1 second, before being presented with the symbol representing the action they are to imagine (<-: "Left", ->: "Right", v: "Legs", +: "Rest"), which lasts on screen for 5500 ms. We record the latter 5000 ms of EEG data. After which, there is a randomized period of rest the lasts between 0.5 and 2 seconds, before the process repeats for another randomly chosen action class. This is done in 4 blocks of 5 trials per action, for a total of 20 trials per action. This procedure is again similar to datasets like BCI Competition 2003 [75], that use non-linguistic cues and randomization of rest / task. At task-time, similar to SSVEP, subjects are told in writing on the screen to perform MI to select a robot skill to execute. Thereafter, a written mapping of a class symbol ({<-, ->, v, +}) to skill ({pick_from_top, pick_from_side, ...}) is presented, and we begin recording EEG Data after a 2-second delay. For muscle tension, there is also a calibration phase, similar to MI, which entails collecting three 500ms-long trials for each class ("Rest", and "Clench") at the start of each experiment. The cues are written on the screen in words. At task time, when appropriate, written prompts are also presented on the screen (e.g. "clench if incorrect"), followed by a written countdown, after which the user has a 500ms window to clench (or not).

## Appendix 6: Decoding Algorithms Details

For both SSVEP and MI, we select a subset of channels and discard the signals from the rest, as shown in Figure 6. They correspond to the visual cortex for SSVEP, and the motor and visual areas for MI (with peripheral areas). For muscle tension (jaw clenching), we retain all channels.

**SSVEP.** To predict the object of interest, we apply Canonical Correlation Analysis (CCA) as shown in [77] to the collected SSVEP data. As each potential object of interest is flashing at a different frequency, we are able to generate reference signals $Y_{f_n}$ for each frequency $f_n$:

$$Y_{f_n} = \begin{bmatrix} \sin(2\pi f_n t) \\ \cos(2\pi f_n t) \\ \sin(4\pi f_n t) \\ \cos(4\pi f_n t) \end{bmatrix}, \ t = \begin{bmatrix} \frac{1}{f_s} & \frac{2}{f_s} & \dots & \frac{N_s}{f_s} \end{bmatrix} \tag{1}$$

where $f_s$ is the sampling frequency and $N_s$ is the number of samples.

Let $X$ refer to the collected SSVEP data, and $Y$ refer to a set of reference signals for a given frequency. The linear combinations of $X$ and $Y$ can be represented as $x = X^\top W_x$ and $y = Y^\top W_y$, and CCA finds the weights $W_x$ and $W_y$ that maximizes the correlation between $x$ and $y$ by solving the following equation:

$$\max_{W_x, W_y} \rho(x, y) = \frac{\mathbb{E}(W_x^\top XY^\top W_y)}{\sqrt{\mathbb{E}(W_x^\top XX^\top W_x)\mathbb{E}(W_y^\top YY^\top W_y)}} \tag{2}$$

By calculating the maximum correlation $\rho_{f_n}$ for each frequency $f_n$ used for potential objects of interest, we are then able to predict the output class by finding $\text{argmax}_{f_n}(\rho_{f_n})$ and matching the result to the object of interest with that frequency.

Furthermore, we are able to return a list of predicted objects of interest in descending order of likelihood by matching each object to a list of descending maximum correlations $\rho_{f_n}$.

**Motor imagery.** To perform MI classification, we first band-pass filter the data between 8Hz - 30Hz, as that is the frequency range that includes the $\mu$-band and $\beta$-band signals relevant to MI. The data is then transformed using the Common Spatial Pattern (CSP) algorithm. CSP is a linear transformation technique that applies a rotation to the data to orthogonalize the components where the over-timestep variance of the data differs the most across classes. We can then use the log-variance of each time series after rotation as features and perform QDA. Thereafter, we extract features by taking the normalized variance of this transformed data (called "CSP-space data"). We then perform Quadratic Discriminant Analysis (QDA) on this data. To calculate our calibration accuracy, we perform K-fold cross validation with $K_{\text{CV}} = 4$, but we use the entire calibrate dataset to fit the classifier for deployment at task-time.

CSP can be very briefly described as a process which orthogonalizes variance. To illustrate in the 2-class case, suppose the $i$-th calibration EEG time-series for class $k$ can be written as $X_k^{(i)} \in \mathbb{R}^{C \times T}$, where $C$ = number of channels, $T$ = number of time-steps, $i \in [1, 20]$, and $k \in \{1, 2\}$. Suppose further that the data is mean-normalized. Then:

$$\hat{\text{Cov}}(X_k) = \frac{1}{20} \sum_{i=1}^{20} \text{Cov}\left(X_k^{(i)}\right)$$
$$= \frac{1}{20} \sum_{i=1}^{20} \frac{1}{T} X_k^{(i)} X_k^{(i)\top} \tag{3}$$

And we perform a simultaneous diagonalization of $\{\hat{\text{Cov}}(X_k)\}$: $\text{Cov}(X_2)^{-1}\text{Cov}(X_1) = Q\Lambda Q^\top$. The transformation of any time-series $X$ into the CSP-space is simply:

$$X_{\text{CSP}} = XQ \tag{4}$$

Note that we only keep the first $N_{\text{CSP}} = 4$ columns of $XQ$. The Python `mne` package [78] provides a multi-class generalization of this algorithm that we use. Feature extraction can be readily done by taking the component-wise variance of $X_{\text{CSP}}$, but we find that taking the normalized component-wise log-variance is better, as corroborated by previous studies [79]:

$$f_p(X) = \log\left(\frac{\text{Var}(X_{\text{CSP},p})}{\sum_{i=j}^{N_{\text{CSP}}} \text{Var}(X_{\text{CSP},j})}\right) \tag{5}$$
$$f(X) = (f_1(X), ..., f_{N_{\text{CSP}}}(X)) \tag{6}$$

where $X_{\text{CSP},j}$ denotes the $j$-th column of $X_{\text{CSP}}$. The QDA step is straightforward: given our calibration dataset $\{f(X_k^{(i)})\}$, we simply fit a quadratic discriminant using the Python `sklearn` package, which allows us to recover a list of MI class predictions in decreasing order of likelihood.

**Muscle tension.** To detect jaw clenches, electromyography (EMG) data is relevant. This is also picked up by our EEG net, and which, for succinctness, we will refer to as EEG data as well.

| | |
|---|---|
| Input dimension | 2048 |
| Number of hidden layers | 5 |
| Hidden layer dimension | 1024 |
| Output dimension | 1024 |
| Number of epochs | 100 |
| Batch size | 40 |
| Optimizer | Adam |
| Learning rate | 0.001 |

Table 7: Feature embedding model and training hyperparameters for object and skill learning.

Facial muscle tension results in a very significant high-variance signal across almost all channels that is very detectable using simple variance-based threshold filters without having to perform any frequency filters. Recall that we record three 500ms-long trials for each class ("Rest", "Clench"). In short, for each of the calibration time-series, we take the variance of the channel with the median variance; call this variance $m$. Then, we just take the mid-point between the maximum $m$ between the rest samples, and the minimum $m$ between the clench samples, and have this be our threshold variance level.

Let $X_k^{(i)} \in \mathbb{R}^{C \times T}$, where $C$ = number of channels, $T$ = number of time-steps, $i \in [1, 3]$, and $k \in \{\text{Rest, Clench}\}$.

$$m_k^{(i)} = \underset{c}{\text{median}} \left\{ \text{Var}\left( X_{k,c}^{(i)} \right) \right\}, \ c \in [1, C] \tag{7}$$

where $X_{k,c}^{(i)}$ denotes the $c$-th row of $X_k^{(i)}$.

$$\text{Threshold} = \frac{1}{2} \left( \max_i \{m_{\text{Rest}}^{(i)}\} + \min_i \{m_{\text{Clench}}^{(i)}\} \right) \tag{8}$$

## Appendix 7: Robot Learning Algorithm Details

### Object and skill learning details

We utilize pre-trained R3M as the feature extractor. Our training procedure aims to learn a latent representation of an input image for inferring the correct object-skill pair in the given scene. The feature embedding model is a fully-connected neural network that further encodes the outputs of the foundation model. Model parameters and training hyperparameters are summarized in Table 7. Collecting human data using a BRI system is expensive. To enable few-shot learning, the feature embedding model is trained using a triplet loss [80], which operates on three input vectors: anchor, positive (with the same label as the anchor), and negative (with a different label). Triplet loss pulls inputs with the same labels together by penalizing their distance in the latent space, as well as pushes inputs with different labels away. The loss function is defined as:

$$\mathcal{J}(a, p, n) = \max \left( \|f(a) - f(p)\|_2 - \|f(a) - f(n)\|_2 + \alpha, \ 0 \right) \tag{9}$$

where $a$ is the anchor vector, $p$ the positive vector, $n$ the negative vector, $f$ the model, and $\alpha = 1$ is our separation margin.

**Generalization test set.** We test our algorithm in the following generalization settings:

*Position and pose.* For position generalization, we randomize the initial positions of all objects in the scene with fixed orientation and collect 20 different trajectories. For the pose generalization, we randomize both the initial positions and orientations of all objects.

*Context.* The context-level generalization refers to placing the target object in different environments, defined by different backgrounds, object orientations, and the inclusion of different objects in the scene. We collect 20 different trajectories with these variations.

*Instance.* The instance generalization aims to assess the model's capability to generalize across different types of objects present in the scene. For our target task (MakePasta), we collect 20 trajectories with 20 different kinds of pasta with different shapes, sizes, and colors.

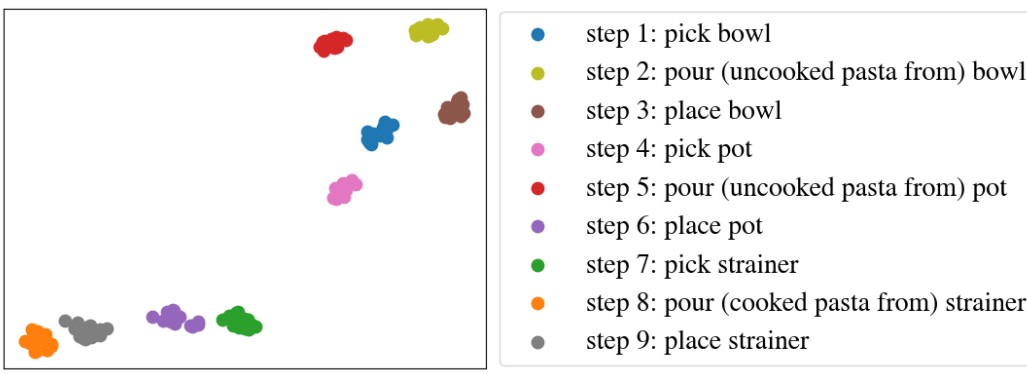

Figure 7: t-SNE visualization of latent representation generated by object and skill learning embedding model for `MakePasta` task pose generalization dataset.

**Latent representation visualization.** To understand the separability of latent representations generated by our object-skill learning model, we visualize the 1024-dimensional final image representations using the t-SNE data visualization technique [81]. Results for the `MakePasta` pose generalization test set are shown in Figure 7. The model can well separate each of the different stages of the task, allowing us to retrieve the correct object-skill pair for an unseen image.

**One-shot parameter learning details**

**Design choices.** We empirically found that using DINOv2's ViT-B model, alongside a 75x100 feature map and a 3x3 sliding window, with cosine similarity as the distance metric, resulted in the best performance for our image resolutions.

**Generalization test set** We test the generalization ability of our algorithm on 1008 unique training and test pairs, encompassing four types of generalizations including 8 position trials, 8 orientation trials, 32 context trials, and 960 instance trials.

*Position and orientation* The position and orientation generalizations, shown in Fig. 8 and Fig. 9 respectively, are tested in isolation, e.g. when the position is varied, the orientation is kept the same.

*Context* The context-level generalization, shown in Fig. 10, refers to placing the target object in different environments, e.g. the training image might show the target object in the kitchen while the test image shows the target object in a workspace. Here, we allow for position and orientation to vary as well.

*Instance* To test our algorithm's capability of instance-level generalization, shown in Fig. 11, we collected a set of four different object categories, each containing five unique object instances. Our object categories consist of mug, pen, bottle, and medicine bottle, whereas the bottle and medicine bottle categories consist of images from both the top-down and side views. We test all permutations within each object category including train and test pairs with different camera views. Here, we allow for position and orientation to vary as well.

**Test set for comparing our method against baselines** We test our method against baselines on 1080 unique training and test pairs, encompassing four types of generalizations including 8 position trials, 8 orientation trials, 32 context trials, 960 instance trials, 48 trials where we vary all four generalizations simultaneously, and 24 trials from the `SetTable` task.

*Position, orientation, context, and instance simultaneously.* Finally, we test our algorithm's ability to generalize when all four variables differ between the training and test image, shown in Fig. 12. Here, the only object category we use is a mug.

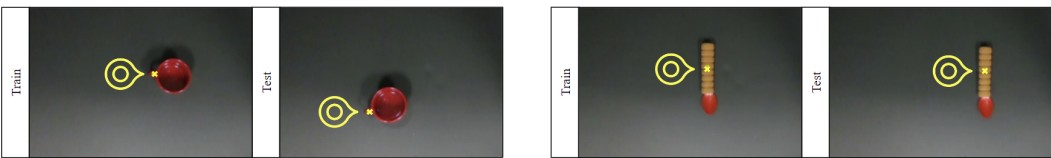

Figure 8: Position generalization. The first train parameter is set on the mug handle. The second train parameter is set on the spoon grip.

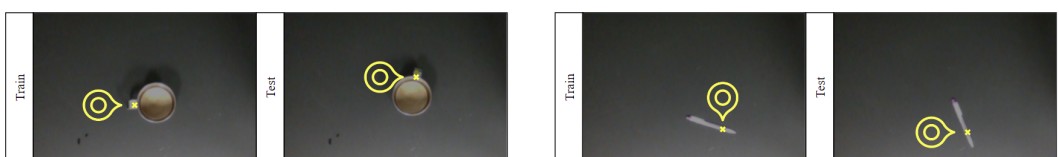

Figure 9: Orientation generalization. The first train parameter is set on the mug handle. The second train parameter is set on the pen grip.

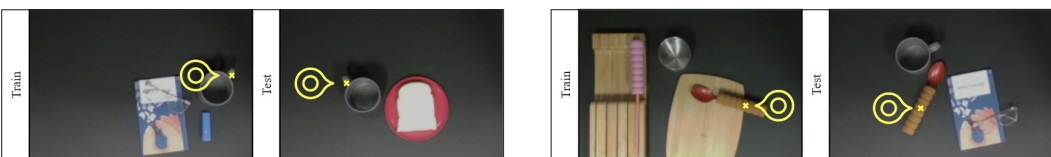

Figure 10: Context generalization. The first train parameter is set on the mug handle. The second train parameter is set on the spoon grip.

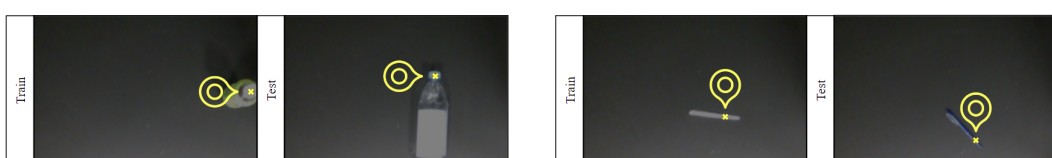

Figure 11: Instance generalization. First pair shows instance generalization with different camera views: from the top and from the side. The first train parameter is set on the bottle cap. The second train parameter is set on the pen grip.

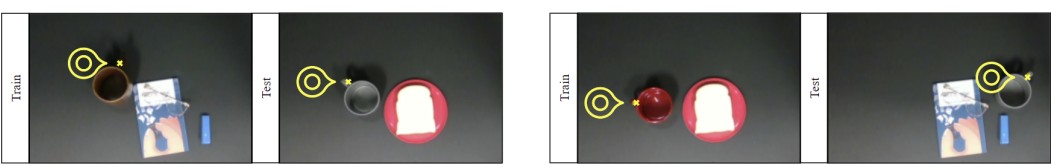

Figure 12: Position, orientation, instance, and context generalization. Both train parameters are set on the mug handle.

| Sukiyaki | | | WipeSpill | | |
|---|---|---|---|---|---|
| Initial | | Goal | Initial | | Goal |

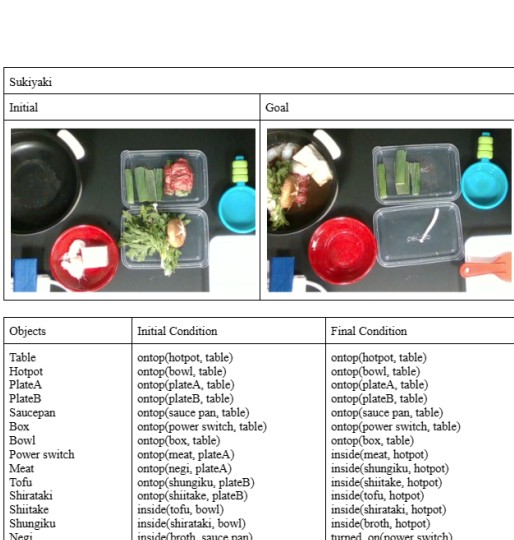 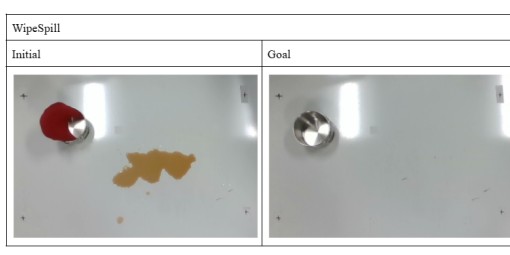

| Objects | Initial Condition | Final Condition |
|---|---|---|
| Table | ontop(hotpot, table) | ontop(hotpot, table) |
| Hotpot | ontop(bowl, table) | ontop(bowl, table) |
| PlateA | ontop(plateA, table) | ontop(plateA, table) |
| PlateB | ontop(plateB, table) | ontop(plateB, table) |
| Saucepan | ontop(sauce pan, table) | ontop(sauce pan, table) |
| Box | ontop(power switch, table) | ontop(power switch, table) |
| Bowl | ontop(box, table) | ontop(box, table) |
| Power switch | ontop(meat, plateA) | inside(meat, hotpot) |
| Meat | ontop(negi, plateA) | inside(shungiku, hotpot) |
| Tofu | ontop(shungiku, plateB) | inside(shiitake, hotpot) |
| Shirataki | ontop(shiitake, plateB) | inside(tofu, hotpot) |
| Shiitake | inside(tofu, bowl) | inside(shirataki, hotpot) |
| Shungiku | inside(shirataki, bowl) | inside(broth, hotpot) |
| Negi | inside(broth, sauce pan) | turned_on(power switch) |
| Broth | turned_off(power switch) | inside(spatula, box) |
| Spatula | in_hand(spatula) | |

| Objects | Initial Condition | Final Condition |
|---|---|---|
| Table | ontop(cup, table) | ontop(cup, table) |
| Cup | inside(napkins, cup) | not_covered(tabletop, liquid) |
| Napkins | covered(tabletop, liquid) | |
| Liquid | | |

| CollectToy | | | SweepTrash | | |
|---|---|---|---|---|---|
| Initial | | Goal | Initial | | Goal |

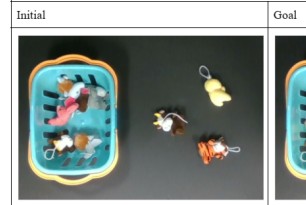 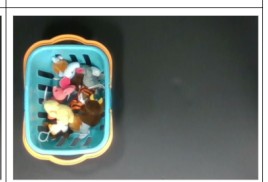 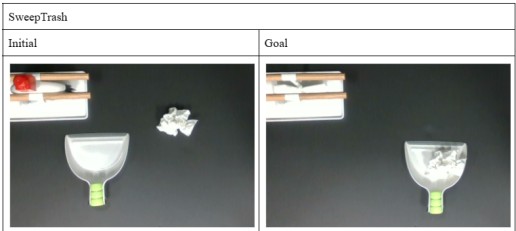

| Objects | Initial Condition | Final Condition |
|---|---|---|
| Table | ontop(basket, table) | ontop(basket, table) |
| Basket | ontop(ToyA~C, table) | inside(ToyA~C, basket) |
| Toys | inside(Toys, basket) | inside(Toys, basket) |
| ToyA~C | | |

| Objects | Initial Condition | Final Condition |
|---|---|---|
| Table | ontop(broom, table) | ontop(dustpan, table) |
| Dustpan | ontop(trash, table) | in(trash, dustpan) |
| Broom | ontop(dustpan, table) | |
| Trash | | |

| CleanBook | | | IronCloth | | |
|---|---|---|---|---|---|
| Initial | | Goal | Initial | | Goal |

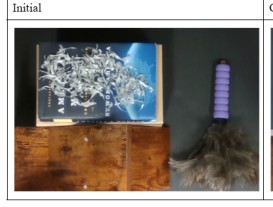 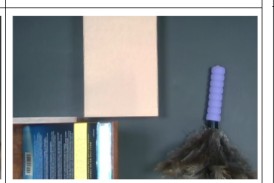 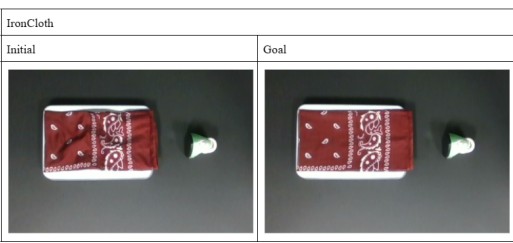

| Objects | Initial Condition | Final Condition |
|---|---|---|
| Table | ontop(book shelf, table) | ontop(book shelf, table) |
| Book shelf | ontop(featherduster, table) | ontop(book, book shelf) |
| Book | ontop(book, table) | not_covered(book, dust) |
| Dust | covered(book, dust) | |
| Featherduster | | |

| Objects | Initial Condition | Final Condition |
|---|---|---|
| Table | ontop(ironing board, table) | ontop(ironing board, table) |
| Iron | ontop(iron, table) | ontop(cloth, ironing board) |
| Cloth | ontop(cloth, ironing board) | not_covered(cloth, wrinkles) |
| Ironing board | covered(cloth, wrinkles) | |

| OpenBasket | | | PourTea | | |
|---|---|---|---|---|---|
| Initial | | Goal | Initial | | Goal |

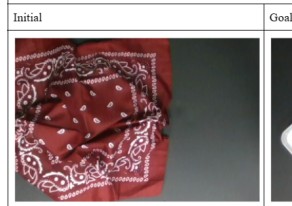 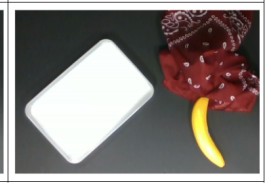

| Objects | Initial Condition | Final Condition |
|---|---|---|
| Table | ontop(basket, table) | ontop(cloth, table) |
| Basket | inside(banana, basket) | ontop(basket, table) |
| Cloth | ontop(cloth, basket) | ontop(banana, table) |
| Banana | | |

| Objects | Initial Condition | Final Condition |
|---|---|---|
| Table | ontop(cup, table) | ontop(cup, table) |
| Cup | ontop(plate, table) | ontop(plate, table) |
| Teabag | ontop(teabag, plate) | inside(teabag, teapot) |
| Teapot | ontop(teapot, table) | inside(tea, cup) |
| Water | inside(water, teapot) | |
| Tea | | |
| Plate | | |

| SetTable | | |
|---|---|---|
| Initial | | Goal |

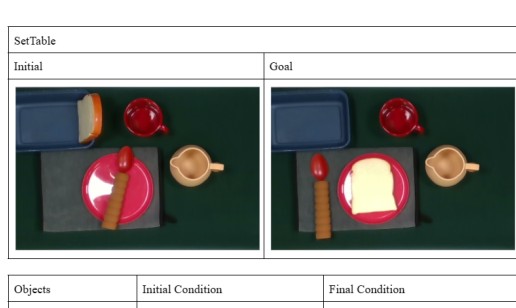

| Objects | Initial Condition | Final Condition |
|---|---|---|
| Table | ontop(lunch mat, table) | ontop(lunch mat, table) |
| Lunch mat | ontop(container, table) | ontop(container, table) |
| Container | ontop(teapot, table) | ontop(plate, lunch mat) |
| Bread | ontop(plate, lunch mat) | ontop(cup, lunch mat) |
| Teapot | ontop(cup, table) | ontop(spoon, lunch mat) |
| Bowl | ontop(spoon, plate) | ontop(bread, plate) |
| Cup | inside(bread, container) | ontop(cup, lunch mat) |
| Spoon | inside(water, teapot) | inside(water, cup) |
| Water | | |

| GrateCheese | | |
|---|---|---|
| Initial | | Goal |

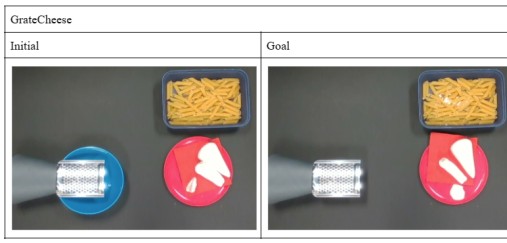

| Objects | Initial Condition | Final Condition |
|---|---|---|
| Table | ontop(plate, table) | ontop(container, table) |
| Container | ontop(cheese block, plate) | inside(pasta, container) |
| Grater | ontop(container, table) | ontop(grated cheese, pasta) |
| Bowl | inside(pasta, container) | |
| Cheese block | ontop(bowl, table) | |
| Grated cheese | ontop(grater, bowl) | |
| Plate | absent(grated cheese) | |
| Pasta | | |

| CutBanana | | |
|---|---|---|
| Initial | | Goal |

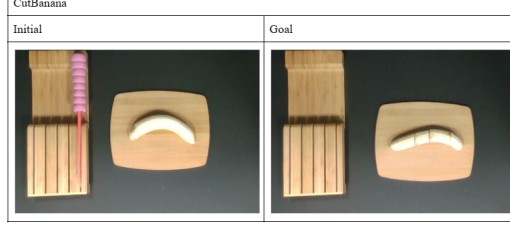

| Objects | Initial Condition | Final Condition |
|---|---|---|
| Table | ontop(cutting board, table) | ontop(cutting board, table) |
| Cutting board | ontop(knife holder, table) | ontop(knife holder, table) |
| Banana | ontop(banana, cutting board) | ontop(banana, cutting board) |
| Knife | inserted(knife, knife holder) | sliced**(banana) |
| Knife holder | not_sliced*(banana) | |

*not_sliced = has no cuts, **sliced = has 3 parallel cuts

| CookPasta | | |
|---|---|---|
| Initial | | Goal |

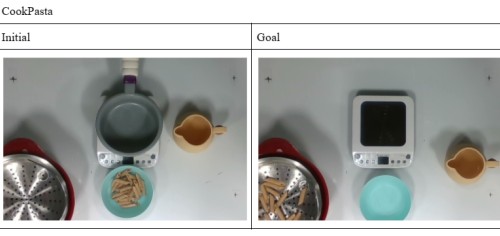

| Objects | Initial Condition | Final Condition |
|---|---|---|
| Table | ontop(stove, table) | ontop(stove, table) |
| Pasta | ontop(pot, stove) | ontop(bowlS, table) |
| BowlS | ontop(bowlS, table) | ontop(bowlL, table) |
| BowlL | ontop(bowlL, table) | ontop(strainer, bowlL) |
| Stove | ontop(strainer, bowlL) | inside(pasta, strainer) |
| Pot | inside(pasta, bowlS) | cooked(pasta) |
| Strainer | inside(water, pitcher) | |
| Water | not_cooked(pasta) | |
| Pitcher | | |

| Sandwich | | |
|---|---|---|
| Initial | | Goal |

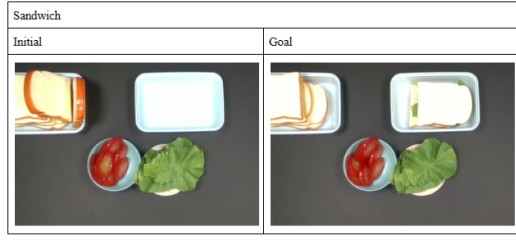

| Objects | Initial Condition | Final Condition |
|---|---|---|
| Table | ontop(containerA, table) | ontop(containerB, table) |
| ContainerA | ontop(containerB, table) | ontop(plate, table) |
| ContainerB | ontop(plate, table) | inside(bread1, containerB) |
| Lid | ontop(tomato, plate) | inside(tomato, containerB) |
| Bread1 | ontop(lettuce, plate) | inside(lettuce, containerB) |
| Bread2 | inside(bread1, containerA) | inside(bread2, containerB) |
| Tomato | inside(bread2, containerA) | ontop(tomato, bread1) |
| Lettuce | | ontop(lettuce, tomato) |
| Plate | | ontop(bread2, lettuce) |

| Hockey | | |
|---|---|---|
| Initial | | Goal |

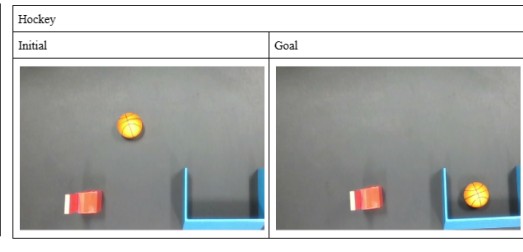

| Objects | Initial Condition | Final Condition |
|---|---|---|
| Table | ontop(ball, table) | ontop(goal, table) |
| Ball | ontop(tool, table) | in(ball, goal) |
| Goal | ontop(goal, table) | |
| Tool | not_in(ball, goal) | |

| OpenGift | | |
|---|---|---|
| Initial | | Goal |

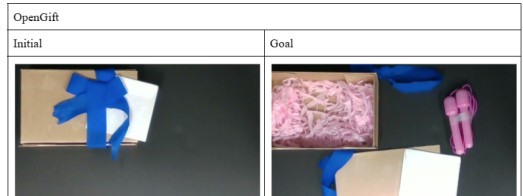

| Objects | Initial Condition | Final Condition |
|---|---|---|
| Table | ontop(gift box, table) | ontop(gift, table) |
| Gift box | ontop(lid, gift box) | |
| Lid | attached(card, lid) | |
| Card | tied(ribbon, gift box) | |
| Ribbon | tied(ribbon, lid) | |
| Gift | tied(ribbon, card) | |
| | inside(gift, gift box) | |

| TicTacToe | | |
|---|---|---|
| Initial | | Goal |

| Objects | Initial Condition | Final Condition |
|---|---|---|
| Table | ontop(marker, table) | filled**(squares1~9) OR gameset***() |
| Eraser | empty*(squares2~9) | **Followed by:** |
| Squares1~9 | | empty*(squares1~9) |
| Marker | | |
| "O" | | |
| "X" | | |

*empty = no "O" or "X" drawn, **filled = exactly 1 "O" or "X" drawn, ***there is a winner

| TrashDisposal | | | |
|---|---|---|---|
| Initial | | Goal | |

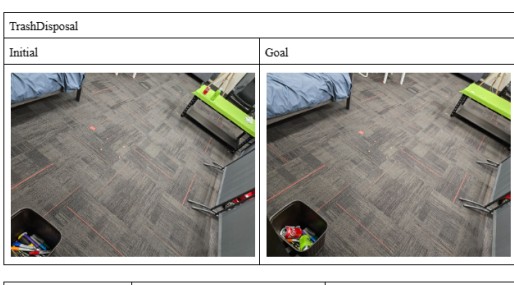

| Objects | Initial Condition | Final Condition |
|---|---|---|
| Table | ontop(can, table) | inside(can, trashcan) |
| Trashcan | ontop(bottle, table) | inside(bottle, trashcan) |
| Can | ontop(trashcan, floor) | |
| Bottle | inroom(table) | |
| | inroom(trashcan) | |

| WaterPlant | | | |
|---|---|---|---|
| Initial | | Goal | |

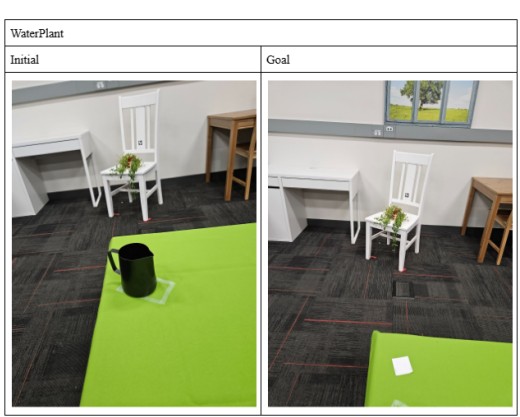

| Objects | Initial Condition | Final Condition |
|---|---|---|
| Table | ontop(bottle, table) | not_dry(plant) |
| Pitcher | ontop(pitcher, table) | dry(chair) |
| Bottle | ontop(plant, chair) | dry(floor) |
| Plant | inroom(chair) | |
| Chair | inroom(table) | |
| | dry(plant) | |
| | dry(chair) | |
| | dry(floor) | |

| CovidCare | | | |
|---|---|---|---|
| Initial | | Goal | |

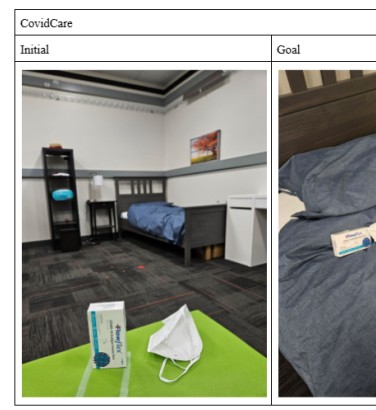

| Objects | Initial Condition | Final Condition |
|---|---|---|
| Table | ontop(mask, table) | ontop(mask, bed) |
| Mask | ontop(testkit, table) | ontop(testkit, bed) |
| Testkit | inroom(table) | |
| Bed | inroom(bed) | |

| PetDog | | | |
|---|---|---|---|
| Initial | | Goal | |

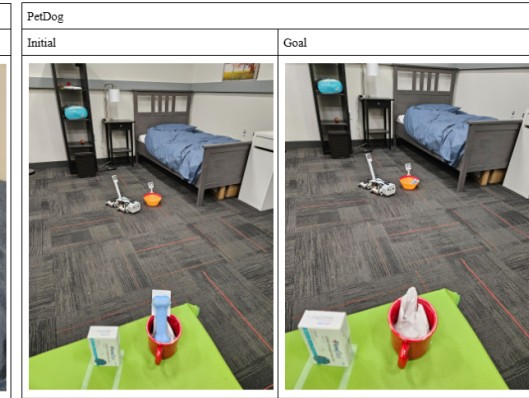

| Objects | Initial Condition | Final Condition |
|---|---|---|
| Table | inroom(table) | inside(bone, bowl) |
| Bowl | inroom(bowl) | petted(dog) |
| Dog | inroom(dog) | |
| Dogbone | ontop(dogbone, table) | |
| Testkit | ontop(testkit, table) | |
| | not_petted(dog) | |

