# OpenReview forum: "NOIR: Neural Signal Operated Intelligent Robots for Everyday Activities"
_robot-learning.org/CoRL/2023/Conference — CoRL 2023 Poster_

### Official Review · Reviewer_SXKL · 2023-07-18

**Confidence:** 3
**Originality:** Good
**Technical Quality:** Good
**Clarity Of Presentation:** Very Good
**Impact:** 4

**Recommendation:**

Weak Accept: I recommend accepting the paper, but will not argue for my recommendation if the majority of other reviewers have a different opinion.

**Review:**

The paper in my opinion is well written and straightforward to understand. The line of argument behind the approach is reasonable and the design decisions made by the authors are well explained and reasonable as well. The approach is evaluated in various experiments and the results seem to be convincing. Citations seem to be adequate, however I am not an expert in this field and therefore can't really tell if important related work has not been cited.

For the skill classification part, If I understood it correctly, the operator can choose from 4 different classes: left hand, right hand, legs, rest. However, I don't see how those 4 classes are connected to all the 20 tasks which are performed by the subjects. There are tasks where for instance multiple/different sub-tasks can be performed with the left hand. Do you show the operator a list of subtasks and how they are associated to the 4 skill classes? This should be clarified in the paper.

I was wondering how the object selection method would perform against a simpler but maybe more robust method like highlighting the objects one after the other and letting the operator select an object based on a simple detected signal pattern (e.g., highlight the objects one after the other until the operator selects the current object by emitting a certain signal). This would basically require a sequence of binary classifications instead of a single N-class classification. In general, a comparison against one or multiple baseline methods for the individual what/how/where parts would have made the paper stronger.

**Quality Of The Limitations Section:**

Limitations are addressed clearly

**Questions For Rebuttal:**

As written above, could the authors please clarify how the 4 skill classes are mapped to a subtask? If multiple actions can be performed with the same hand on an object, how do you choose between the corresponding actions?

**Robotics Focus:**

Sufficient demonstration on hardware

**Summary Of Paper:**

The authors present an approach where an operator can teleoperate a robot via an EEG-based Brain-Computer-Interface. The main contribution in my opinion is a learning-based method which allows the system to preselect the intended action of the operator (e.g., pick mug from the left side) which reduces the need of the operator to interact with the system through EEG and therefore speeds up task execution. The authors decompose the "human goal decoding" into three parts. 1) What object 2) How to interact (what task) and 3) Where to interact (task parameters). For each part they developed a different Brain-Computer-Interface suited for selecting choices or choosing parameters.

**Summary Of Recommendation:**

I chose my rating mainly because I did not find any major flaws in the paper.

---

### Official Review · Reviewer_KKsW · 2023-07-20

**Confidence:** 3
**Originality:** Fair
**Technical Quality:** Very Good
**Clarity Of Presentation:** Very Good
**Impact:** 2

**Recommendation:**

Weak Reject: I recommend rejecting the paper, but will not argue for my recommendation if the majority of other reviewers have a different opinion.

**Review:**

Strengths:
* The paper is well organized and seems to provide sufficient information to the reader. Therefore, the clarity of the paper is high.

Weaknesses:
* The main contribution of the paper, the versatile task execution using a robotic agent through brain signals, appears to be performed without leveraging robot learning methods. The object and skill selection and parameter learning methods were only applied to a few tasks to show their effectiveness. Thus, the quality and significance of the paper in terms of robot learning is not high.
* Related work on BRI systems using the object and skill selection and parameter learning methods are not mentioned in the paper. Thus, it is difficult to assess the originality of the paper with regard to robot learning.


**Quality Of The Limitations Section:**

Limitations are addressed clearly

**Questions For Rebuttal:**

* I would like to know the difference between the proposed system and existing BRI systems from the robot learning point of view.

**Robotics Focus:**

Sufficient demonstration on hardware

**Summary Of Paper:**

The main idea and contributions of the paper are the following points:
* The authors developed a brain-robot interface (BRI) system that is capable of performing a wide variety of tasks while adapting to user preferences.
* The authors successfully demonstrated that the proposed BRI system could perform a wide variety of 20 household activities, such as making Sukiyaki, cleaning a book and watering a plant, using a robotic agent through brain signals.


**Summary Of Recommendation:**

The contribution of the paper is not sufficient: weakness in both quality and significance of the paper. The main contribution of the paper seems to be less related to robot learning.

---

### Official Review · Reviewer_beZJ · 2023-07-20

**Confidence:** 4
**Originality:** Good
**Technical Quality:** Fair
**Clarity Of Presentation:** Fair
**Impact:** 3

**Recommendation:**

Weak Accept: I recommend accepting the paper, but will not argue for my recommendation if the majority of other reviewers have a different opinion.

**Review:**

This paper deals with the very challenging topic of controlling a robot from a person's brain waves.
Therefore, the importance of this research is very high.
However, extensive expertise is required to understand this paper.
For example, brain, BIM, robotics, and machine learning.

The experiments were primarily a detailed analysis of the proposed system. Due to the nature of the research, comparative validation with other similar methods is expected to be complicated.


**Quality Of The Limitations Section:**

Limitations are not well addressed

**Questions For Rebuttal:**

My concern in controlling robots from brain waves is safety.
Safety is an essential aspect for both people and robots.
Does the person always need to wear the EEG to control the robot?
Is there any danger of the robot destroying objects around it?
Another aspect that seems essential is versatility. Are there any individual differences in this system?
Does a person need to be trained in using this system to pass away the robot?
The above may be slightly different from the purpose and contribution of this paper.
However, if there is a direction for a solution to these questions, they should be coordinated.

In order to control a robot from brain waves, a mechanism will be needed better to capture the characteristics of the brain wave data.
Several machine learning methods have been selected for the proposed method, but it is unclear why they are considered suitable.
In addition, the novelty of machine learning techniques such as robot learning is unclear.
The proposed system appears only to use existing models.

The text in figures in the text is very small and difficult to read.

Many references to the Appendix. Therefore, it is difficult to understand this paper alone fully.
The study is rich in content, and the omission of many details may be a disadvantage.

**Robotics Focus:**

Relevant but unlikely to deploy to hardware in near future

**Summary Of Paper:**

This paper proposes a framework for codifying robots from human EEG data called NOIR.
NOIR uses machine learning methods to analyze human EEG data acquired from EEG and performs human goal decoding.
The robot then performs primitive skills along with human goal prediction.
A benchmark dataset was used in the experiment. The performance of the proposed system was tested on various skills.

**Summary Of Recommendation:**

This paper is challenging to read due to the field's broad scope.
However, it is interesting because it makes use of recent machine-learning techniques.

---

### Official Review · Reviewer_r85X · 2023-07-20

**Confidence:** 4
**Originality:** Good
**Technical Quality:** Very Good
**Clarity Of Presentation:** Very Good
**Impact:** 3

**Recommendation:**

Weak Accept: I recommend accepting the paper, but will not argue for my recommendation if the majority of other reviewers have a different opinion.

**Review:**

# Strengths
- The paper is very presented very clearly and thoughtfully. In particular, the authors have done a really good job of making the text digestible to an audience who may not have much familiarity with BRI/BCI techniques (e.g. Question and Answer section provided in Appendix, etc.)
- NOIR is demonstrated on a truly challenging set of long-horizon tasks. The activities selected for evaluating the system are convincingly difficult and well-motivated.

# Weaknesses
- The human study presented is very small, making it difficult to extrapolate much from the results.


**Quality Of The Limitations Section:**

Additional details required

**Questions For Rebuttal:**

1. Can the authors expand on how a task attempt is defined? Does this mean that a participant reached an unrecoverable state at some point in task execution and the task had to be reset from the beginning? Or does this indicate that they had to reattempt a particular skill if it did not yield the expected outcome? Some clarity here would help with the interpretation of Table 1.

2. Can the authors provide some insights into why some tasks seem to require more attempts? Are participants encountering decoding errors (which may be mitigated by the safety mechanism but maybe not avoided completely?) or do they require more challenging/creative manipulation that caused the participant to need to try different strategies? I'm hoping to get a more intuitive understanding of how difficult these tasks are relative to one another. How do I interpret long task times and few attempts on one task vs. the same task time and many attempts on another?

3. I don't see human time, presented as a percentage of total time in Table 1, as an intuitive metric. Human time is ~80 for most tasks, except for TrashDisposal and CovidCare - why is it lower for these two tasks? There seems to be some ambiguity as to whether human time (as a percentage of total time) is lower because decision-making/decoding occurred faster or if the robot skill execution actually took longer. Instead, I think it would be helpful to understand how much raw time humans are spending on decision-making/decoding for each manipulation step (for selecting how and where the robot will interact with an object). That way, we can better compare how long this process takes between tasks.

4. The human study sample size is quite small, which significantly limits the conclusions that can be made about the effectiveness of the system. The paper would benefit from additional experiments with more participants, though it is understandable if this is not possible to execute within a short rebuttal period. At the very least, without these experiments, it would be helpful for the authors to more explicitly address this as a limitation in the text of the paper. While a small statement about the potential universality of the system is provided in the appendix, I would argue that a statement addressing the size of the study should more materially address how the reader can/should understand the actual evaluation results and should appear in the actual body of the main paper.

5. I think the paper would benefit from more clearly differentiating itself from existing BRI literature. While it is clear that NOIR enables control for a wide variety of complex tasks, it would be helpful to illustrate just how challenging the demonstrated tasks are than those in previous literature (require more variety of skills? require more skills to be used in succession? are harder to plan for? etc.). I think it would also be interesting to compare/contrast this work to non-invasive BRI that enable continuous real-time (or hybrid) control, especially since there is evidence that users prefer to be actively involved in task execution rather than simply selecting goals before the robot moves.


**Robotics Focus:**

Sufficient demonstration on hardware

**Summary Of Paper:**

This work proposes a BRI system that allows users to control a robot to perform a diverse set of tasks using only non-invasive EEG. Users command the robot to complete tasks by first selecting the object to interact with, then selecting the type of interaction (corresponding to a particular robot action primitive), and finally selecting where to interact with the object. The system leverages robot learning to reduce human effort when selecting both how and where to interact with objects, attempting to predict users' intent based on the object they are manipulating. In a human study with 3 participants, the authors demonstrate their method on a set of 20 complex tasks.

**Summary Of Recommendation:**

The paper is clear and its central ideas are thoughtfully communicated. While the authors could benefit from more strongly differentiating this work from the existing literature from a technical perspective, their contributions are convincing overall. Further, the authors showcase their method on the most challenging set of long-horizon tasks that this reviewer is personally familiar with, which is impressive in and of itself. This paper is limited most by its extremely small human study size, which makes it difficult to accept the results at face value. While n < 5 studies are not uncommon in BCI/BRI literature, this study presumably did not face the same types of participant recruitment challenges since this work was performed exclusively with able-bodied people. This paper appears sound in every other aspect, however, so I will recommend that the paper be accepted on the condition that other reviewers do not also view the limited study size as a severe limitation.

---

### Author Response · Authors · 2023-08-14
**General discussion**

We thank all the reviewers for their thoughtful comments and questions. We are delighted to see that most of you find our research problem important, robotic tasks challenging, and presentation accessible.

1. Reviewers request a more comprehensive literature review on the brain-robot interface. We have added a new related work section comparing our work with representative, recently published EEG-based BRI works, as summarized here:
||Paper | Num of tasks | Example task | Brain signal | Real / Simulation | Use of machine learning / robot learning||
|---|:---:|:---:|:---:|:---:|:---:|:---:|----|
||Ours | 20 | making sukiyaki | object, skill, parameter | real | for decoding & learning human preferences of objects, skills, and parameters|
||[1] | 1 | navigation | evaluative feedback | sim | using decoded reward to train RL agent|
||[2] | 2 | navigation / reaching | evaluative feedback | sim | using decoded reward to train RL agent|
||[3] | 1 | clean table | object | real | no learning|
||[4] | 1 | binary object selection | evaluative feedback | real | no learning|
||[5] | 1 | reaching | evaluative feedback | real | using decoded reward to train RL agent|
||[6] | 1 | navigation | motor command | sim | for decoding|
||[7] | 1 | grasping | motor command | real | for decoding|
||[8] | 3 | grasping | motor command | real | no learning|
||[9] | 5 | reaching and grasping | motor command | real | for decoding|

Many earlier BRI works are summarized in [10]. As shown, previous works are limited to a few simple tasks such as reaching and grasping. We want to emphasize that our work is the first in many perspectives, and this research area does not have well-defined benchmarks or frameworks. Hence it is difficult to directly compare our system with existing ones since our system enables us to perform long-horizon manipulation tasks with a much wider range of objects. It is also the first work incorporating machine learning and robot learning techniques in all the major components of the system.


2. There were also concerns about the human study sample size. We had only three participants but they performed all the tasks. Each subject took about 20+ hours to complete all the tasks (spreading across multiple days), hence we cannot recruit a lot of subjects. The rationale was that EEG is known for its generality across humans, however, EEG-based BRI systems were limited to a few simple tasks – hence we want to emphasize that our system can solve more tasks instead of more humans.
Nonetheless, we recruited **additional** 6 human subjects (2 females and 4 males) and asked them to perform the OpenBasket task. They all can successfully perform the task. Their performance, in terms of task horizon, attempts, completion time, and human time percentage is similar to what is reported in the paper (Table 1). This indicates that our system is effective and consistent among different human participants.  Due to limited time, we will continue the experiments and report their results for the rest of the tasks in the future.

|||Subj4| Subj5|Subj6|Subj7|Subj8|Subj9|Average(4-9)|Original Average(1-3)||
|---|:---:|:---:|:---:|:---:|:---:|:---:|:---:|:---:|:---:|----|
||Task horizon| 4.00| 4.00|7.00|4.00|4.00|5.00|4.67|5.33|
||# Attempts	| 2.00	|1.00	|2.00	|2.00	|1.00	|3.00	|1.83|	1.67|
||Total time (min)	|16.28	|14.48	|25.07	|12.67	|15.93	|15.61	|16.68	|15.90|
||Human time (min)	|14.69	|12.37	|21.58	|10.54	|13.90	|13.50	|14.43	|13.04|
||Manipulation time (min)	|1.59	|2.12	|3.50	|2.14	|2.04	|2.11	|2.25	|2.86|
||Human time percentage (%)|	90.24	|85.39	|86.05	|83.15	|87.20	|86.48	|86.42	|82.03|


[1] Akinola, Iretiayo, et al. Accelerated robot learning via human brain signals. ICRA, 2020.

[2] Wang, Zizhao, et al. Maximizing BCI human feedback using active learning. IROS, 2020.

[3] Akinola, Iretiayo, et al. Task level hierarchical system for BCI-enabled shared autonomy. IEEE-RAS 17th International Conference on Humanoid Robotics (Humanoids) 2017.

[4] Salazar-Gomez, Andres F., et al. Correcting robot mistakes in real time using EEG signals. ICRA 2017.

[5] Schiatti, Lucia, et al. Human in the loop of robot learning: Eeg-based reward signal for target identification and reaching task. ICRA 2018.

[6] Aljalal, Majid, et al. Robot navigation using a brain computer interface based on motor imagery. JMBE 2019.

[7] Xu, Yang, et al. Shared control of a robotic arm using non-invasive brain–computer interface and computer vision guidance. RAS 2019.

[8] Chen, Xiaogang, et al. Control of a 7-DOF robotic arm system with an SSVEP-based BCI. IJNS 2018.

[9] Meng, Jianjun, et al Noninvasive electroencephalogram based control of a robotic arm for reach and grasp tasks. Scientific Reports 2016.

[10] Aljalal, Majid, et al. Comprehensive review on brain-controlled mobile robots and robotic arms based on electroencephalography signals. Intelligent service robotics 2020.

---

### Decision · Program_Chairs · 2023-08-30

**Decision:**

Accept (Poster)

**Comment:**

The reviewers commend the paper for being well-written and organized. The proposed system to teleoperate a robot via an EEG-based Brain-Computer-Interface with a real human-study evaluation was well received. I encourage the authors to further address any remaining points raised in the reviews for the final version of the paper.